

# Improved transfer learning using textural features conflation and dynamically fine-tuned layers

Raphael Ngigi Wanjiku[1], Lawrence Nderu[2] and Michael Kimwele[2]

[1] Nexford University, Washington DC, United States
[2] Computing, Jomo Kenyatta University of Agriculture and Technology, Nairobi, Kenya

## ABSTRACT

Transfer learning involves using previously learnt knowledge of a model task in addressing another task. However, this process works well when the tasks are closely related. It is, therefore, important to select data points that are closely relevant to the previous task and fine-tune the suitable pre-trained model's layers for effective transfer. This work utilises the least divergent textural features of the target datasets and pre-trained model's layers, minimising the lost knowledge during the transfer learning process. This study extends previous works on selecting data points with good textural features and dynamically selected layers using divergence measures by combining them into one model pipeline. Five pre-trained models are used: ResNet50, DenseNet169, InceptionV3, VGG16 and MobileNetV2 on nine datasets: CIFAR-10, CIFAR-100, MNIST, Fashion-MNIST, Stanford Dogs, Caltech 256, ISIC 2016, ChestX-ray8 and MIT Indoor Scenes. Experimental results show that data points with lower textural feature divergence and layers with more positive weights give better accuracy than other data points and layers. The data points with lower divergence give an average improvement of 3.54% to 6.75%, while the layers improve by 2.42% to 13.04% for the CIFAR-100 dataset. Combining the two methods gives an extra accuracy improvement of 1.56%. This combined approach shows that data points with lower divergence from the source dataset samples can lead to a better adaptation for the target task. The results also demonstrate that selecting layers with more positive weights reduces instances of trial and error in selecting fine-tuning layers for pre-trained models.

## INTRODUCTION

Transfer learning has made deep learning implementation easier and adaptable in many industries. The process involves reusing knowledge from a previous pre-trained model's task in another domain's task and dataset. Studies suggest this process works well in closely related tasks. For example, using a pre-trained model with learnt knowledge of car images to classify lorry images instead of using it to classify flowers. This example results in positive transfer learning due to sharable features from both domains: cars and lorries.

Corresponding author
Raphael Ngigi Wanjiku,
phaelgi@gmail.com

However, if the target dataset were different, for example, flowers, it could lead to negative transfer learning due to differences in the sharable features.

Transfer learning is broadly classified based on labels and feature spaces. The label classification is categorised into transductive (where the label information comes from the source domain and the tasks are in similar domains), inductive (where the label information comes from the target domain) and unsupervised transfer learning, where neither of the domains is labelled. The feature space classification is categorised into homogeneous (where both tasks belong to the same domain, although they may have different marginal distributions) and heterogeneous (where the source and target tasks are from different domains). These two classifications can further be explored based on instances, features, parameters and relationships between the domains as outlined by *Zhuang et al. (2021)* and *Zhang & Gao (2022)*. Furthermore, recent views look into the combinations of data, data properties and models. These views include instance-based, network-based, mapping-based and adversarial-based (*Liang, Fu & Yi, 2019*). Other views of transfer learning include instances (semi-supervised), multi-view, multitask learning with lifelong learning, transitive learning and reinforcement learning, which are promising for future transfer learning approaches (*Choi et al., 2018*; *Zhang & Gao, 2022*; *Lu et al., 2020*).

The transfer learning uses several methods: fine-tuning last-k layers (*Mingsheng et al., 2017*; *Xuhong, Grandvalet & Davoine, 2020*), freezing initial layers, and feature extraction. However, the selection of data points and regularisation has recently been introduced (*Li et al., 2020*). This study combines a selection of data points and fine-tuning of selected layers for positive transfer.

Deep learning involves the use of many hidden layers in neural networks. Convolutional neural networks (CNN) are among the neural networks that have accelerated the adoption of deep learning. Most pre-trained models using CNNs have been trained on the ImageNet dataset—with over fourteen million images (*ImageNet, 2021*). The hidden layers in deep neural networks can learn complex patterns and behaviours supporting supervised and unsupervised learning. This learning support has allowed model developers to develop many transfer learning models, such as those hosted on TensorFlow Hub (*TensorFlow Hub, 2023*). These pre-trained models can then be adopted on a need basis; for most transfer learning involving image data, ImageNet pre-trained models provide a better solution to training from scratch (*Sun et al., 2022*). However, reusing these models requires careful consideration to avoid negative knowledge transfer (*Wu et al., 2021*).

## Research contributions

This study proposes an alternative method for improving the pre-trained model's performance in two stages: selecting relevant target data points and dynamic layer selection using Kullback–Leibler divergence ($D_{KL}$). The method combines the pre-trained models' performance improvement at the two stages giving an overall model performance.

The contributions of this study are:

- The pre-trained model on the target tasks achieves better transfer learning adaptation by selecting higher-quality data points in the target dataset through textural divergence

measurements. The selected data points can be used in other model improvement tasks like augmentation. The data points selection builds better performance confidence in their application in machine learning algorithms.

- By selecting layers with higher positive weights, the pre-trained model can fine-tune much better, which improves its performance. These positive weights can give faster convergence, improving the adaptability of the target tasks.

- Pre-trained models can improve performance and user confidence in the target tasks during transfer learning processes by preparing the modelling pipeline using quality target dataset samples and adaptable-fine-tuned layers. This approach results in an effective and efficient pre-trained model selection procedure; from the previous trial and error selection of pre-trained models and fine-tuning layers.

## Textural features of image data

In image processing, texture refers to an objective function representing the brightness and intensity variation of an image's pixels (*Tuceryan & Jain, 1993*). Texture explains images' smoothness, roughness, regularity and coarseness, as noted by *Laleh & Shervan (2019)*. Texture gives the sequential illumination patterns of the pixels in an image and the image grey tones in the pixel's neighbourhood (*Dixit & Hegde, 2013*). Textural features in an image can be classified into three: low-level, mid-level and high-level. The classification is based on pixel levels, image descriptors and image data representation (*Bolón-Canedo & Remeseiro, 2019*). The features in an image are analysed to understand the spatial arrangement of the pixels' grey tones after their extraction. The extraction process can be categorised according to transformation, structure, model, graphical, statistical, entropy and learning views. The low-level image features are heavily used in image classification, utilising colour, texture and shape attributes. These attributes are passed through filters, quantified using statistical descriptors such as entropy and correlation, and ranked through relevance indices.

The two commonly used methods in textural analysis are the grey-level co-occurrence matrix (GLCM) and the local binary pattern (LBP) (*Ershad, 2012*). The GLCM was introduced by *Haralick, Shanmugam & Dinstein (1973)* to represent the pixels' brightness levels using a matrix that combines the grey levels intervals, direction and amplitude change. The GLCM descriptor has 14 features, with interval distance and orientation being the most important (*Andrearczyk & Whelan, 2016*). This study evaluates three features: correlation, homogeneity and energy. The LBP uses the local textural patterns in an image and compares the pixel's neighbouring grey levels. The comparison of the neighbours uses representative binary numbers described using histograms. The LBP is a robust textural descriptor used in edge detection and textural description (*Zeebaree et al., 2020*).

## Textural features conflation of image data

Conflation refers to a merge of two or more probability distributions. The concept was introduced by *Hill (2011)*. Given probability $P_1,\ldots,P_n$, the conflation(&) of 1 to $n$ is expressed in Eq. (1) below, with Eqs. (2) and (3) giving the conflation for discrete

distributions. Equation (4) shows the conflation of continuous distributions for textural features in the source and target domain data points.

$$Q = \&(P_1, \ldots, P_n) \tag{1}$$

where $Q$ is the merged probability distribution.

$$\&(P_1, \ldots, P_n) = \frac{f_1(x)f_2(x), \ldots, f_n(x)}{\sum_y f_1(y)f_2(y), \ldots, f_n(y)} \tag{2}$$

where $f_1, \ldots, f_n$ refers to probability density functions of the textural features. This equation can be rewritten as;

$$\&(SP_1, \ldots, SP_n) = \frac{Sf_1(x)Sf_2(x), \ldots, Sf_n(x)}{\sum_y Sf_1(y)Sf_2(y), \ldots, Sf_n(y)} \tag{3}$$

$$\&(TP_1, \ldots, TP_n) = \frac{Tf_1(x)Tf_2(x), \ldots, Tf_n(x)}{\int_{-\infty}^{\infty} Tf_1(y)Tf_2(y), \ldots, Tf_n(y)} \tag{4}$$

where $SP_1, \ldots, SP_n$ refers to the probability distributions of the source domain samples. The probability distributions of the target domain samples can be represented using Eq. (4) by substituting S with T.

The conflation of features removes redundant features resulting in a balanced probability distribution (*Hill, 2011*).

## Model layer fine-tuning in transfer learning

Fine-tuning is one of the methods of transfer learning. The process involves selecting training layers and freezing the weights of the pre-trained model on a target task. In most cases, the first layers of a model are chosen due to their ability to extract features as opposed to the last layers, which are mainly used for classification purposes (*Coskun et al., 2017*). Fine-tuning has been a manual process involving selecting the first or the initial layers (in most cases, the last three layers of the network), as reported in the literature (*Deniz et al., 2018*; *Fan, Lee & Lee, 2021*). *Vrbančič & Podgorelec (2020)* noted that models have specific architectures that sometimes make their layer selection inefficient and fine-tuning a trial-and-error process.

## RELATED WORK

This proposed work builds on two previous works by *Wanjiku, Nderu & Kimwele (2022)*. The first work looks into selecting relevant data points using textural features. The authors use three datasets: Caltech 256, Stanford Dogs 120 and MIT Indoor Scenes on two pre-trained models—VGG16 and MobileNetV2. The proposed approach adds datasets and models, extending the process by dynamically selecting fine-tunable layers in the transfer learning process. The authors use four datasets in the second study: CIFAR-10, CIFAR-100, MNIST, and Fashion-MNIST on six pre-trained models. In the second study, the authors evaluate the selection of pre-trained models' layers based on weights. They use cosine similarity and later $D_{KL}$ on the cosine similarity. The proposed work only looks at the $D_{KL}$ divergence and further utilises the data points previously identified in the

processing pipeline. In this study, two smaller datasets are added to validate the model: one of the use cases of transfer learning is in cases of limited datasets.

The selection of data points in transfer learning has been documented in various literature. In a study by *Weifeng & Yizhou (2017)*, data points with similar low-level features are identified and selected in the target domain to address the insufficient data using Gabor filters. The feature selection in this proposed study uses pre-trained CNN layer filters, while Ge & Yu (2017) used Gabor filters. Their work describes the features using histograms, while the proposed study uses conflated probability distributions. The two studies also differ through their datasets and pre-trained models: the researchers used three pre-trained models (AlexNet, VGG-19 and GoogleNet) and three datasets (Caltech 256, MIT Indoor Scenes and Stanford Dogs 120), while the proposed uses five pre-trained models (ResNet50, DenseNet169, VGG-16, InceptionV3 and MobileNetV2) and six additional datasets.

*Zhuang et al. (2015)* compare the features between the source and target domains from a generative adversarial network (GAN) using Kullback–Leibler divergence on the features' probability distributions. The distributions formation utilises a temperature-softmax function which controls the samples used in the source domain. The use of the Kullback–Leibler divergence and temperature softmax function is also done in this research. The two studies differ in the pre-trained models and datasets used, where the researchers utilise the ResNet architecture and one dataset. In contrast, the proposed uses four additional architectures and nine datasets to validate the conflation method.

*Luo et al. (2018)* utilise an optimal similarity graph to select low-level features in video semantic recognition. The researchers use semi-supervised learning to address the curse of dimensionality preventing information loss between video pairs while acquiring the features of the local structure. This approach differs from the proposed method in feature extraction (the proposed uses convolution layers and conflates the features) and dynamic fine-tuning. In contrast, the researchers use an optimal similarity graph. However, the two methods use divergence measures in comparing the low-level features.

*Gan, Singh & Joshi (2017)* address the conflation of probability distributions intending to understand the semantics in the text strings utilising long short-term memory recurrent neural network (LSTM-RNN) in business analytics for entity profiles. The conflation method has also been used in geographic information systems (GIS) in merging geospatial datasets (*De Smith, Goodchild & Longley, 2018*), in the detection of robotic activities (*Rahman et al., 2021*), and in features dimensionality reduction involving large datasets as researched by *Mitra, Saha & Hasanuzzaman (2020)*.

*Royer & Lampert (2020)* introduce the flex-tuning method. The method proposes fine-tuning a single layer while freezing the rest of the model. This process is iterated until a group of best unit layers is selected for use in the final transfer learning process. The researchers' approach differs from the proposed method based on the layers' selection criteria. The researchers select the weights based on the layer that performs better, while the proposed method selects the layer based on its positive weights. However, both methods consider the weights in the layer selection process. Furthermore, the proposed method integrates the selection of quality data points aiding overall network performance.

*Yunhui et al. (2019)* introduce the SpotTune method that uses Gumbel-Softmax sampling on two ResNet architectures. In the method, a decision policy determines the best layers to be selected through a lightweight network that evaluates each instance. This approach differs from the proposed approach: the researchers use a lightweight network in layer selection using two ResNets and five datasets, while the proposed uses weights in layer selection, five pre-trained models, and nine datasets.

Other layer selection studies have considered evolution algorithms. *Satsuki, Shin & Hajime (2020)* utilise the genetic algorithm where genotypes (representing the layer weight) with the highest accuracy are selected for fine-tuning. The researchers improve the algorithm by using the tournament selection algorithm, experimenting with three datasets: SVHN21, Food-101 and CIFAR-100. *Vrbančič & Podgorelec (2020)* introduce the differential evolution algorithm that selects and represents the pre-trained model's layers using binary values. All the selected layers are assigned a binary value of 1.

All these documented layer selection methods have been evaluated on one pre-trained model and one or two datasets, which is insufficient and needs more evaluation. However, the proposed method uses more models and datasets. Furthermore, the feature extraction in the first phase of the model is done using the convolutional layers of the same pre-trained model to be used in the transfer learning process.

Apart from selecting features, transfer learning is used in various industries, including biomedical, manufacturing, and deep learning model security. In medical imaging, numerous issues affect the data used in deep learning, including legal and ethical issues which limit the data size and the acquisition expense. *Matsoukas et al. (2022)* demonstrate the effectiveness of feature reuse in the early layers and weight statistics when using transfer learning. The researchers demonstrate that it is advantageous when using vision transformers (ViTs) through the feature reuse gain in transfer learning since they do not have the available inducted bias in CNNs. The adaptation noted in medical datasets flows from weights learned from the ImageNet and the extracted low-level features in the pre-trained models. The researchers use four datasets: APTOS2019, CBIS-DDSM, ISIC 2019, and CHEXPERT on four models: two ViT models (DEIT and SWIN) and two CNN models (RESNETs and INCEPTION). They conclude that feature reuse plays a critical role in effective transfer learning, with the early layers showing a strong feature reuse dependence. Their work differs from the proposed method on the number of pre-trained models and datasets.

*Mabrouk et al. (2022)* use transfer learning on three medical datasets—ISIC-2016, PH2, and Blood-Cell datasets to improve the Internet of Medical Things (IoMT) performance in melanoma and leukemia. Transfer learning extracts the image features while the chaos game optimisation selects the good features. In a skin classification task by *Rodrigues et al. (2020)*, transfer learning is used on skin lesions, typical nevi and Melanoma using IoT systems. The researchers use VGG, Inception, ResNets, Inception-ResNet, Xception, MobileNet and NASNet as pre-trained models, applying SVM, Bayes, RF, KNN, and MLP classifiers. The researchers experiment with the method on two datasets: ISIC and PH2. Their classification study aimed to address issues faced by medical teams during lesion classification. These issues include using various sizes and lesions shapes, the patient's skin

colour, personnel experience, and fatigue on the classification day. The work by these researchers uses more datasets and pre-trained models but differs from the proposed method on feature conflation and dynamic layer selection.

*Duggani et al. (2023)* develop a hybrid transfer learning model from two pre-trained CNN models to improve classification for melanoma. In the study, they predict the fine-grained differences in skin lesions on the skin surface. The features extracted from the two models are concatenated, and an SVM classifier is added at the end of the model. The concatenation improves the accuracy performance values of the traditional CNN models on the ISBI 2016 dataset. The researchers used AlexNet, GoogleNet, VGG16, VGG19, ResNet 18, ResNet50, ResNet101, ShuffleNet, MobileNet, and DenseNet201 as the pre-trained models. *Nguyen et al. (2022)* have also documented skin lesions classification using the Task Agnostic Transfer Learning (TATL). The researchers concatenated the extracted features while this work conflates them, selecting the ones to evaluate the target task samples.

Transfer learning has been used further in medical imaging to classify other diseases. *Chouhan et al. (2020)* utilise five pre-trained models to classify pneumonia images. *Niu et al. (2021)* classify COVID-19 lung CT images using the distant domain transfer learning (DDTL) model on three source domain datasets (unlabeled Office-31, Caltech-256, and chest X-ray) and one target domain dataset (labelled COVID-19 lung CT). Their study aims to reduce the distribution shift between the domain data. *Zoetmulder et al. (2022)* use CNN pre-trained models on three brain T1 brain segmentation tasks: MS lesions, brain anatomy, and stroke lesions using natural images and T1 brain MRI images. *Raza et al. (2023)* use transfer learning to classify and segment Alzheimer's disease on the brain's grey matter images. *Holderrieth, Smith & Peng (2022)* use transfer learning to address the technical variability of MRI scanners and the differences in subject populations on the UK Biobank MRI data (three datasets) focusing on age and sex attributes. In all these medical imaging cases, *Kim, Cosa-Linan & Santhanam (2022)* note that the most common transfer learning models in medical scenarios include—AlexNet, ResNet, VGGNet and GoogleNet since they can be easily customised.

In fault-tolerant systems, *Li et al. (2020)* use transfer learning to address the limited data available in these systems. The researchers use simulation data on convolutional neural network architecture integrating domain adaptation techniques. The developed model is deployed on a pulp mill plant and a continuously stirred tank reactor. *Nawar et al. (2023)* use transfer learning to optimise power generation planning and bill savings potential. Their Building-to-building transfer learning model uses the deep learning—transformer model in forecasting power savings. The researchers evaluated the algorithm on a large commercial building using LSTM and RNN and concluded that the transformer model performed better than the LSTM and RNN architectures. In satellite data applications, researchers tap into massive-dataset-trained foundation models such as ImageNet and GPT-3 to improve the performance of downstream tasks in different satellite application domains.

In their work, *Simumba & Tatsubori (2023)* use foundation models by allowing pre-trained model weights in cases of various input channels. Using weights helps the

downstream applications address the difference between satellite data and computer vision models. The researchers test the approach on precipitation data-trained models. Transfer learning has also been used in human activity recognition (HAR) systems: *An et al. (2023)* use pre-trained models trained with offline HAR classifiers on new users. The researchers introduce representation analysis for transferring specific features from the offline users while maintaining a good complexity analysis in the target setting. *Sharma et al. (2023)* attempt to recognise human behaviour from real-time video. The researchers classify the behaviour as suspicious or usual using data from the real-time video frames on the Novel 2D CNN, VGG16, and ResNet50 pre-trained models.

In a recent study by *Mehta & Krichene (2023)*, transfer learning enhances deep learning model security. The security of private models is paramount to protecting deep learning models that are plausible for bad actors to attack, revealing information from the training examples (differential privacy). The researchers propose using transfer learning as a promising technique for improving the accuracy of private models. The process involves training a model on a dataset with no privacy concerns and then privately fine-tuning it on a more sensitive dataset. The researchers simulate the adjustments on the ImageNet-1k, CIFAR-100, and CIFAR-10 datasets.

## METHODOLOGY

### Proposed study's approach

The proposed method comprises two parts: the selection of quality dataset samples and the dynamic selection of the pre-trained model's fine-tunable layers, as shown in Fig. 1.

As shown in Fig. 1, the target and source domain data samples are compared based on their textural features resulting in a final target dataset as expressed in Eqs. (5) to (8). The pre-trained model layers are then selected for transfer learning to accomplish a target CNN task as shown in Eqs. (9) to (13).

### Selection of quality image dataset

The selection of quality images involves extracting textural features from a target domain image, conflating its textural features' probability distributions, and comparing the resultant distribution with the conflated distributions in the source domain images.

**Definition 1**. The conflation of a target image's textural features. Given a target image $T_{i1}$, its textural features $T_{if1}, \ldots, T_{ifn}$, the conflated probability distribution can be expressed in Eq. (5) as;

$$\&(T_{if1}, \ldots, T_{ifn}) = \frac{T_{if1}(x)T_{if2}(x), \ldots, T_{ifn}(x)}{\int_{-\infty}^{\infty} T_{if1}(y)T_{if2}(y), \ldots, T_{ifn}(y)} \tag{5}$$

where $T_{if}$ represents a target image's feature. We can proceed and equate the conflated value to $\&T_{if}$ as expressed in Eq. (6);

$$(\&T_{if}) = \&(T_{if1}, \ldots, T_{ifn}) \tag{6}$$

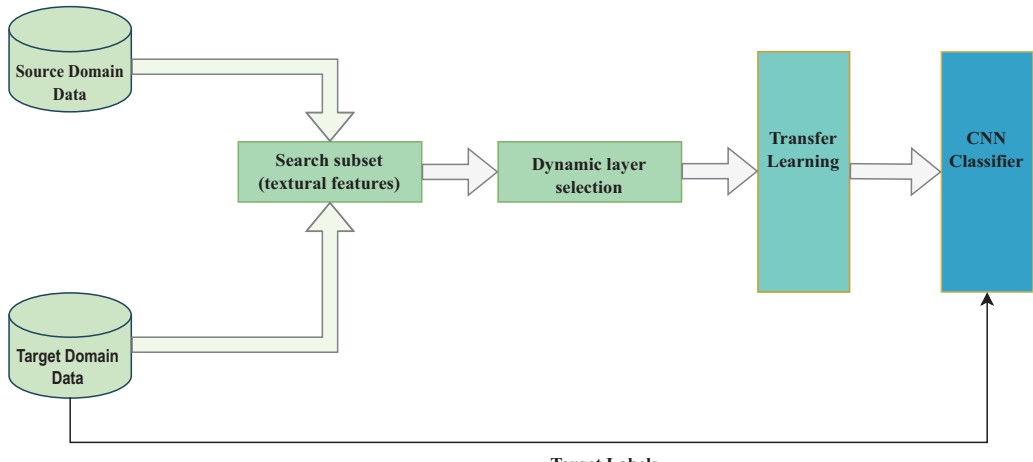

**Figure 1** Study's conceptual framework.  

**Definition 2.** The conflation of a source image's textural features. Given a source image $S_{i1}$, its textural feature vectors $S_{if1}, \ldots, S_{ifn}$ distributions can be conflated into $\&S_{if}$ as expressed in Eq. (7) below;

$$\&(S_{if}) = \frac{S_{if1}(x)S_{if2}(x), \ldots, S_{ifn}(x)}{\int_{-\infty}^{\infty} S_{if1}(y)S_{if2}(y), \ldots, S_{ifn}(y)} \tag{7}$$

where $S_{if1}$ represents a probability distribution of the first feature in a selected source image feature.

Once the source conflated distributions are identified, a vector of the source image conflated probability distribution is used to check its divergence from the target image conflated distribution, as shown in Eq. (8).

$$D_{KL}\big((\&T_{if}) \parallel (S_{if})\big) = \int_{-\infty}^{+\infty} (\&T_{if})(x) log \frac{(\&T_{if})(x)}{\&(\&S_{if})(x)} dx \tag{8}$$

where $D_{KL}$ represents Kullback–Leibler divergence.

Finally, we select images whose $D_{KL}$ is lower than the average $D_{KL}$ of the source image distributions.

## Selection of fine-tunable layers in pre-trained models

The fine-tunable layers selection involves identifying convolutional layers in a pre-trained model, their positive and negative weights and selecting fine-tunable layers by utilising the weights' divergence.

**Definition 3**. Identification of a convolutional layer. Given a pre-trained model $M$, a convolutional layer $C_l$ is expressed as follows;

$$C_l = \begin{cases} M_l, & if \quad M_{ln} = \text{``}conv\text{''} \\ otherwise \end{cases} \tag{9}$$

where $M_l$ is the model's layer, $M_{ln}$ represents the model's layer name which identifies a convolutional layer if its name contains the keyword "*conv*"; otherwise, the layer is skipped.

**Definition 4**. Identification of positive and negative weights. Given a convolutional layer $C_l$, a weighting filter $C_{lw}$ is a vector of $n \times n$ kernel size. Given two filters, $x$ and $y$, we can reshape the tensors from $C_{lw}^x \in \mathbb{R}^{n*n}$ and $C_{lw}^y \in \mathbb{R}^{n*n}$ into $C_{lw}^{x'}$ and $C_{lw}^{y'}$, respectively. We can express the positive and negative weight units as $C_{lw}^{x'+ve}$ and $C_{lw}^{x'-ve}$. Therefore each weight filter becomes a single tensor of positive and negative weights, as expressed in Eq. (10).

$$C_{lw}^{x'} = Tensor\left(C_{lw1}^{x'+ve}, \ldots, C_{lwn}^{x'-ve}\right) \tag{10}$$

where $C_{lw1}^{x'+ve}$ represents the first positive weight unit of filter $x$ for the convolutional layer $C_l$.

**Definition 5**. Divergence measure between layers. Given two single-dimensional layers $C_{lw}^{x'}$ and $C_{lw}^{y'}$, we can calculate their differences by converting the vectors into probability distributions based on their positive or negative weights vectors. Utilising $D_{KL}$, the divergence of the positive weight vectors is expressed in Eq. (11).

$$D_{KL}\left(p(C_{lw}^{x'+ve}) \parallel p(C_{lw}^{y'+ve})\right) = \int_{-\infty}^{+\infty} p\left(C_{lw}^{x'+ve}\right)(x) log \frac{p\left(C_{lw}^{x'+ve}\right)(x)}{p\left(C_{lw}^{y'+ve}\right)(x)} \, dx \tag{11}$$

where $p$ refers to a probability distribution.

We can simplify this further by substituting $p(C_{lw}^{x'+ve})$ with $p(x)$ and $p(C_{lw}^{y'+ve})$ with $p(y)$, as shown in Eq. (12).

$$D_{KL}(p(x) \parallel p(y)) = \int_{-\infty}^{+\infty} p(x)(x) log \frac{p(x)(x)}{p(y)(x)} \, dx \tag{12}$$

The layers with the lower divergence measures are then selected for use in the fine-tuning process.

## Algorithm

The following steps summarise the proposed approach:

i) Select an image $T_{i1}$ from a target dataset.

ii) Extract $T_{i1}$ textural features using a pre-trained model $M$, giving vectors of features $T_{if1}, \ldots, T_{ifn}$.

iii) Convert the vectors of features into probability distributions $p(T_{if1}), \ldots, p(T_{ifn})$

iv) Conflate the probability distributions ($\&T_{if}$) in (iii) into one probability distribution $p(T_{if})$.

v) Repeat the procedure for the source images.

vi) Calculate the average Kullback–Leibler divergence ($D_{KL}$) of the source images.

vii) Compare the $D_{KL}$ of $T_{i1}, \ldots, T_{in}$ to the average of the source selecting data points whose $D_{KL}$ is lower. These images form the final target dataset.

viii) Identify the convolution layers in pre-trained model $M$ using the keyword "*conv*".

ix) Identify the positive and negative weights in each convolutional layer $C_l$.

x) Create probability distributions of the positive and negative weight filter units.

xi) Calculate the $D_{KL}$ between positive or negative probability distributions between any two layers.

xii) Repeat the procedure in (xi).

xiii) Select the layers with the lowest $D_{KL}$ as candidates for fine-tuning.

xiv) Perform transfer learning on the target task utilising the target dataset in step (vii) and the selected layers in step (xiii).

## Divergence measures

Divergence is a measure of the difference between any two probability distributions. This proposed approach uses $D_{KL}$ and compares it to four other divergence measures: Jensen–Shannon, Bhattacharya, Hellinger and Wasserstein.

*Kullback–Leibler*: *Theodoridis (2015)* presents it as a measure between two probability distributions, as expressed in Eq. (12). The divergence gives a zero measure when the two distributions are equal. Adding the zero measure, we can rewrite Eqs. (12) to (13).

$$D_{KL}(p(x) \parallel p(y)) = \int_{-\infty}^{+\infty} p(x)(x) log \frac{p(x)(x)}{p(y)(x)}\, dx \tag{13}$$

where $D_{KL}(p(x) \parallel p(y))$ *if and only if* $x = y$.

*Jensen–Shannon*: A symmetrised version of the $D_{KL}$ that measures the distance between two probability distributions. Unlike the $D_{KL}$, it has high computational costs in search operations (*Nielsen & Nock, 2021*).

*Hellinger distance*: A measure of the difference between any two probability distributions in a shared space. Also known as Jeffrey's distance but has a higher computational complexity than the $D_{KL}$, as noted by *Greegar & Manohar (2015)*.

*Wasserstein distance*: A measure of the difference between any two probability distributions. It uses the concept of moving an amount of earth and the distance involved. The divergence has been used in optimal transportation theory, as *Villani (2003)* noted.

*Bhattacharya*: A measure between two probability distributions that gives the cosine angle to interpret the overlapping angle between them. Like the other mentioned divergence measures, it is highly complex despite its good performance.

The Kullback–Leibler divergence has been used to model the physical microstructure properties of steel (*Lee et al., 2020*), separation of multi-source speech sources (*Togami et al., 2020*), and extracting features in the development of an impulse-noise resistant LBP. *Yuhong et al. (2019)* have used $D_{KL}$ to develop a scale filter bank in a CNN model to create a down-sampled spectrum from two distributions.

## Datasets

This study uses nine publicly available image datasets: CIFAR-10, CIFAR-100 (*Krizhevsky, Nair & Hinton, 2014a, 2014b*), MNIST (*LeCun, Cortes & Burges, 1998*), Fashion-MNIST

(*Xiao, Rasul & Vollgraf, 2017*), Caltech 256 (*Griffin, Holub & Perona, 2022*), Stanford Dogs 120 (*Aditya et al., 2011*), MIT Indoor Scenes (*Ariadna & Antonio, 2009*), ISIC 2016 (*Gutman et al., 2016*) and ChestX-ray8 (*Xiaosong et al., 2017*) to examine the proposed approach. Other studies have used these datasets, making them suitable for performance comparison. The data from the ChestX-ray8 dataset is a subset of over 108,000 images. The smaller ChestX-ray8 and ISIC 2016 sets have been used to show the advantages of transfer learning in cases of inadequate data. Table 1 below shows the dataset sizes and sets.

## Data preparation

Since the approach utilises selected images from the larger datasets, the images are input into the pre-trained models with 224 × 224 pixel dimensions. The images are converted into grayscale, and their features are extracted using the first convolutional layer of the selected model. The pre-trained model is the feature extractor since it is also used in the final transfer learning process. This attribute makes it ideal for preparing the final transfer learning environment. Once the feature selection is made through the proposed approach, the dataset is categorised as a training or test dataset.

## Experimental setup and settings

This study uses five pre-trained models: ResNet50, DenseNet169, InceptionV3, VGG16, and MobileNetV2 (used to show the proposed approach performance on small networks). The inputs to the models have been scaled to 224 × 224 pixels with the InceptionV3 model using an upsampling layer and VGG16 taking 4,096 neurons in the last layer before classification. These pre-trained models have been trained on the ImageNet dataset. The pre-trained models and their parameters are listed in Table 2, with InceptionV3 having the most layers (*Team, 2023*). The study also uses a custom CNN model to show the proposed methods' effects on a non-pre-trained model.

The experiments have been conducted using the TensorFlow framework using the Keras library on the PaperSpace platform (A4000, 45 GB RAM, 8 CPU with 16 GB GPU). The training of the models involved the selected datasets without data augmentation. However, during training, fine-tuned model regularisation was performed using Dropout and Batch normalisation.

## Proposed approach methods

The proposed approach introduces four methods from image textural features and pre-trained model layer selection views.

## Textural features view in image data

This view utilises two methods: Above average $D_{KL}$ and below-average $D_{KL}$.

*Above average* $D_{KL}$: The method uses data points whose $D_{KL}$ is higher than the average for all the data points in their category.

*Below average* $D_{KL}$: The method uses data points whose $D_{KL}$ is lower than the average of other samples in their class.

**Table 1** Study datasets.

| Dataset | Training | Validation | Classes |
|---|---|---|---|
| CIFAR-10 | 50,000 | 10,000 | 10 |
| CIFAR-100 | 50,000 | 10,000 | 100 |
| MNIST | 60,000 | 10,000 | 10 |
| Fashion-MNIST | 60,000 | 10,000 | 10 |
| Caltech 256 | 21,425 | 9,182 | 257 |
| Stanford Dogs 120 | 12,000 | 8,580 | 120 |
| MIT Indoor | 5,360 | 1,340 | 67 |
| ISIC Melanoma | 900 | 379 | 2 |
| ChestX-ray8 | 3,200 | 800 | 4 |

**Table 2** ResNet50 textural features accuracy performance.

| Model | Parameters (millions) | Layers |
|---|---|---|
| ResNet50 | 25,636,712 | 50 layers |
| DenseNet169 | 14,307,880 | 169 layers |
| InceptionV3 | 23,885,392 | 42 layers |
| VGG16 | 138,357,544 | 16 layers |
| MobileNetV2 | 3,500,000 | 53 layers |
| Custom CNN | 106,082 | 12 layers |

## Layer selection view in pre-trained models

This view uses two methods: Positive weights $D_{KL}$ and negative weights $D_{KL}$.

*Positive weight* $D_{KL}$: The method evaluates divergence between two distributions formed from positive weights of filters of a pre-trained model's convolutional layers.

*Negative weight* $D_{KL}$: The method compares two distributions formed from negative weight filters in the pre-trained model's convolutional layers.

$D_{KL}$ is used and compared to four divergence measures in both views to determine the reasons behind its selection.

## Commonly used transfer learning methods

Since the introduced methods in the proposed approach aim to improve the transfer learning process, they are compared with these four commonly used methods in transfer learning:

*Standard fine-tuning*: The method replaces the classification layer with a classification layer for the target task's classes.

*Last-k layer fine-tuning*: It involves replacing the last-k layers in the network with other layers suitable for the target task. These layers can be 1, 2 or 3 in the pre-trained model.

# RESULTS

This section looks at the results of the four methods. It is presented as follows: results on the textural features methods, the dynamic selection, and the commonly used methods and complexities. The results indicate the performance measures using accuracy.

## Results on conflated textural features methods

The two textural methods: GLCM and LBP accuracy performance, are shown in Tables 3–7 for the various datasets and models.

GLCM's energy and homogeneity give the best accuracy performance compared to the LBP and GLCM's correlation. LBP gives the lowest performance compared to the GLCM properties, while ResNet50 performs lowest on CIFAR-100 among the datasets.

In the VGG16 model, correlation and energy perform best among the GLCM properties. The CIFAR-10 give the highest performance across all the properties, while CIFAR-100 still gives the least accuracy.

GLCM's Energy and LBP perform better than the other properties on the InceptionV3 model. GLCM's correlation and homogeneity give the least performance for the InceptionV3, and the CIFAR-10 dataset gives the best accuracy performance across the four textural properties among the datasets. Figure 2A shows the conflated performance of CIFAR-10 samples on the VGG16 pre-trained model.

Figure 2 shows that samples below the average $D_{KL}$ perform better, illustrating the importance of selecting quality data points in a dataset.

GLCM's energy and LBP give better results than the other properties on the DenseNet169 model. GLCM's correlation gives most of the least accuracy performance values, and the CIFAR-100 dataset still gives the least performance.

GLCM's energy and homogeneity give the best accuracies for the MobileNetV2 pre-trained model, while the GLCM's correlation gives the least accuracy. GLCM's energy gives more than half the best results of all four properties, followed by GLCM's homogeneity, LBP and correlation. Figure 3 shows the energy property performance of Fashion-MNIST and MNIST on the MobileNetV2.

The dataset samples with below-average $D_{KL}$ give high accuracy when using both MNIST and Fashion-MNIST, showing the divergence between the data points and their effect in adapting the pre-trained model in the target task. The relevance of these target data points to the source samples is essential to the adaptation.

## Results on dynamically selected layer methods

The dynamic layer selection methods form the second part of the model. The results of the selected layers for the pre-trained model are presented in Tables 8–12.

Before using the selected conflated $D_{KL}$, MNIST performs best when utilising selected dynamic layers, with the CIFAR-100 dataset giving the least performance. However, using conflated $D_{KL}$ samples of the CIFAR-100 dataset and the positive $D_{KL}$ dynamically selected layers improves the ResNet50 pre-trained model's performance, as seen in Table 8. The negative $D_{KL}$ dynamically selected layers also result in minimal improvement due to the conflated dataset samples.

**Table 3  VGG16 textural features accuracy performance.**

| Dataset | ResNet50 | | | | | | | |
| --- | --- | --- | --- | --- | --- | --- | --- | --- |
| | GLCM | | | | | | LBP | |
| | Correlation | | Homogeneity | | Energy | | | |
| | L | H | L | H | L | H | L | H |
| Caltech 256 | 91.02 | 90.68 | 92.14 | 90.68 | 93.14 | 91.68 | 91.36 | 90.84 |
| MIT Indoor | 93.42 | 93.06 | 94.19 | 94.21 | 93.87 | 93.16 | 93.24 | 92.56 |
| Stanford Dogs | 98.24 | 97.34 | 99.12 | 98.85 | 99.42 | 98.59 | 99.34 | 99.02 |
| CIFAR10 | 95.12 | 95.11 | 95.32 | 91.42 | 96.34 | 92.12 | 95.14 | 95.10 |
| CIFAR100 | 35.12 | 32.14 | 32.25 | 31.89 | 32.47 | 32.14 | 33.45 | 32.04 |
| MNIST | 90.48 | 89.02 | 89.24 | 88.34 | 91.53 | 90.52 | 92.04 | 89.36 |
| Fashion MNIST | 82.47 | 82.19 | 83.47 | 83.44 | 83.09 | 82.13 | 82.24 | 82.22 |
| CRX8 | 84.01 | 82.02 | 84.34 | 83.97 | 82.64 | 81.01 | 82.74 | 82.64 |
| Melanoma | 81.34 | 76.50 | 79.63 | 83.47 | 80.23 | 79.25 | 79.36 | 78.96 |

**Table 4  InceptionV3 textural features accuracy performance.**

| Dataset | VGG16 | | | | | | | |
| --- | --- | --- | --- | --- | --- | --- | --- | --- |
| | GLCM | | | | | | LBP | |
| | Correlation | | Homogeneity | | Energy | | | |
| | L | H | L | H | L | H | L | H |
| Caltech 256 | 93.41 | 93.19 | 91.15 | 90.63 | 92.69 | 89.58 | 97.98 | 96.15 |
| MIT Indoor | 97.21 | 95.28 | 90.71 | 88.87 | 90.69 | 89.03 | 90.58 | 90.24 |
| Stanford Dogs | 98.59 | 98.05 | 94.18 | 91.11 | 94.24 | 89.89 | 93.21 | 92.08 |
| CIFAR10 | 96.67 | 92.57 | 97.35 | 95.21 | 96.64 | 95.83 | 96.84 | 96.72 |
| CIFAR100 | 32.04 | 31.59 | 31.97 | 30.99 | 33.28 | 32.87 | 33.24 | 33.17 |
| MNIST | 93.47 | 92.64 | 92.17 | 89.59 | 91.67 | 91.10 | 90.21 | 90.14 |
| Fashion MNIST | 88.54 | 87.76 | 89.47 | 89.34 | 88.64 | 88.14 | 89.24 | 88.35 |
| CRX8 | 72.67 | 69.37 | 72.14 | 68.90 | 79.75 | 69.12 | 74.18 | 72.58 |
| Melanoma | 83.69 | 81.69 | 82.47 | 80.34 | 83.40 | 77.85 | 80.14 | 79.68 |

Using dynamically selected layers performs well compared to the standard methods (compared in Results on Methods against commonly used measures). However, an improvement is noted when employing samples with below-average $D_{KL}$, as seen in Table 9. The CIFAR-100 dataset gives the least accuracy, while the MNIST dataset performs best. Figure 4 shows the performance of the ISIC 2016 dataset on the VGG16 pre-trained model, with and without conflated selected images and dynamic layers.

The dataset with data points below average $D_{KL}$ performs better on VGG16 pre-trained model than those with above-average $D_{KL}$. The performance improves as the dynamically selected layers fine-tune the pre-trained model. In Fig. 5, the selected layer (layer 1) with

**Table 5 DenseNet169 textural features accuracy performance.**

| Dataset | InceptionV3 | | | | | | | |
| --- | --- | --- | --- | --- | --- | --- | --- | --- |
| | GLCM | | | | | | LBP | |
| | Correlation | | Homogeneity | | Energy | | | |
| | L | H | L | H | L | H | L | H |
| Caltech 256 | 91.25 | 91.69 | 91.89 | 90.48 | 98.24 | 95.18 | 90.08 | 90.02 |
| MIT Indoor | 94.05 | 93.68 | 93.41 | 92.99 | 92.45 | 92.14 | 94.36 | 93.79 |
| Stanford Dogs | 89.47 | 86.32 | 90.24 | 90.04 | 89.36 | 88.98 | 91.24 | 90.57 |
| CIFAR10 | 93.47 | 92.11 | 94.78 | 94.68 | 96.71 | 92.94 | 91.84 | 90.14 |
| CIFAR100 | 31.24 | 31.08 | 32.57 | 32.18 | 32.59 | 31.06 | 32.04 | 31.82 |
| MNIST | 91.43 | 90.12 | 90.47 | 90.38 | 93.67 | 87.98 | 92.57 | 92.01 |
| Fashion MNIST | 88.96 | 87.45 | 89.10 | 88.64 | 91.47 | 89.46 | 90.57 | 90.12 |
| CRX8 | 83.20 | 79.78 | 84.15 | 81.20 | 82.45 | 82.40 | 84.69 | 82.58 |
| Melanoma | 83.14 | 82.69 | 82.67 | 81.67 | 84.69 | 83.57 | 80.45 | 79.95 |

**Table 6 MobileNetV2 textural features accuracy performance.**

| Dataset | DenseNet169 | | | | | | | |
| --- | --- | --- | --- | --- | --- | --- | --- | --- |
| | GLCM | | | | | | LBP | |
| | Correlation | | Homogeneity | | Energy | | | |
| | L | H | L | H | L | H | L | H |
| Caltech 256 | 91.31 | 89.63 | 94.14 | 93.24 | 96.12 | 95.12 | 97.14 | 96.12 |
| MIT Indoor | 93.42 | 93.18 | 93.17 | 93.09 | 94.74 | 92.54 | 94.27 | 93.82 |
| Stanford Dogs | 95.24 | 95.29 | 94.36 | 94.02 | 94.14 | 93.45 | 92.43 | 92.37 |
| CIFAR10 | 94.29 | 92.48 | 95.25 | 94.06 | 94.28 | 93.67 | 92.67 | 91.36 |
| CIFAR100 | 33.14 | 32.64 | 32.89 | 32.46 | 34.27 | 31.96 | 32.48 | 31.24 |
| MNIST | 78.36 | 77.32 | 79.05 | 78.36 | 79.51 | 79.24 | 79.39 | 78.49 |
| Fashion MNIST | 90.36 | 89.14 | 90.39 | 90.06 | 91.47 | 90.78 | 91.36 | 90.47 |
| CRX8 | 83.01 | 72.15 | 81.26 | 78.50 | 84.58 | 82.26 | 85.65 | 84.15 |
| Melanoma | 82.47 | 75.92 | 88.69 | 87.13 | 89.56 | 77.62 | 88.24 | 87.01 |

lower $D_{KL}$ is seen to have more excitatory weights (light ones) in the first channel of the layer than layer 9 of the VGG16 model. Its selection coincides with the first layers being able to extract features better.

A similar trend of improvement in fine-tuned dynamically selected layers' model is noted in the InceptionV3 pre-trained model, as seen in Table 10. The MNIST datasets still perform well, probably owing to their size of data points and classes compared to the poorly performing CIFAR-100 dataset. The two smaller datasets: ChestX-ray8 and ISIC 2016, also perform well.

**Table 7 ResNet50 accuracy performance using the selected DKL methods.**

| Dataset | MobileNetV2 | | | | | | | |
|---|---|---|---|---|---|---|---|---|
| | GLCM | | | | | | LBP | |
| | Correlation | | Homogeneity | | Energy | | | |
| | L | H | L | H | L | H | L | H |
| Caltech 256 | 98.59 | 91.52 | 93.99 | 86.58 | 98.55 | 97.54 | 98.72 | 93.95 |
| MIT Indoor | 94.58 | 92.68 | 97.35 | 89.87 | 95.45 | 93.34 | 95.38 | 92.96 |
| Stanford Dogs | 98.38 | 98.22 | 99.29 | 96.57 | 99.15 | 96.95 | 98.76 | 96.34 |
| CIFAR10 | 65.34 | 62.39 | 64.12 | 64.58 | 65.47 | 64.38 | 61.89 | 62.34 |
| CIFAR100 | 31.67 | 30.54 | 32.27 | 31.53 | 33.68 | 32.47 | 33.57 | 32.14 |
| MNIST | 79.25 | 78.64 | 78.36 | 77.21 | 78.49 | 77.98 | 79.52 | 79.01 |
| Fashion MNIST | 91.25 | 90.87 | 90.25 | 88.39 | 91.36 | 90.28 | 91.24 | 90.35 |
| CRX8 | 72.98 | 66.89 | 77.25 | 72.45 | 76.10 | 70.54 | 73.69 | 73.50 |
| Melanoma | 92.80 | 88.70 | 91.90 | 90.52 | 91.56 | 89.90 | 93.56 | 88.80 |

a) CIFAR10 (correlation) on VGG16

b) Caltech256 (LBP) on InceptionV3

**Figure 2 CIFAR10 (correlation) and Caltech256 (LBP) on VGG16 and InceptionV3 pre-trained models.**

The trend of low performance continues with the CIFAR-100 dataset for the DenseNet169 pre-trained model, as seen in Table 11. The MNIST datasets continue to perform best by utilising below-average selected samples, improving the adaptation process's performance in the target task.

The MobileNetV2 pre-trained model gives similar results to the other four pre-trained models, as shown in Table 12. The MNIST dataset gives the best accuracy among the datasets, and a performance improvement is noted when data points of below-average $D_{KL}$ are used together with the positive $D_{KL}$ dynamically selected layers in the transfer learning process. The excellent performance of the positive $D_{KL}$ when using the selected samples is seen in the precision shown in Table 13.
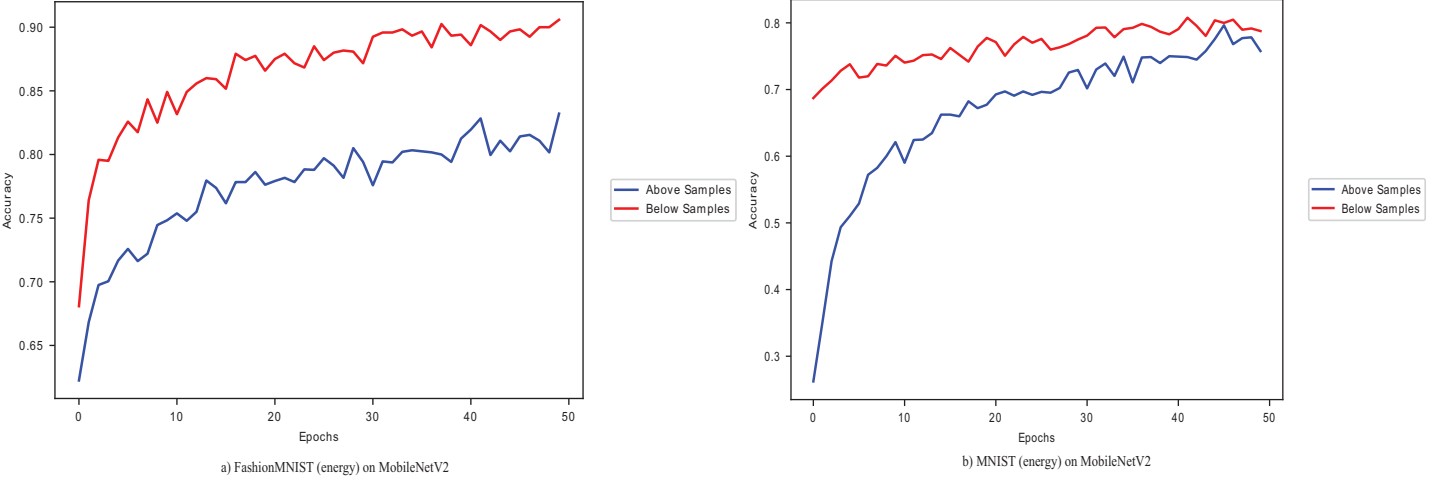

a) FashionMNIST (energy) on MobileNetV2

b) MNIST (energy) on MobileNetV2

**Figure 3** MNIST (energy) and Fashion MNIST (energy) on MobileNetV2.

**Table 8 VGG16 accuracy performance using the selected DKL methods.**

| Dataset | ResNet50 | | | |
| --- | --- | --- | --- | --- |
| | Kullback–Leibler methods | | | |
| | Positive | | Negative | |
| | Before samples | After samples | Before samples | After samples |
| Caltech 256 | 97.04 | 98.15 | 96.23 | 96.25 |
| MIT Indoor | 96.64 | 97.24 | 95.98 | 96.12 |
| Stanford Dogs | 92.16 | 93.01 | 91.06 | 91.77 |
| CIFAR10 | 54.18 | 55.29 | 53.58 | 53.89 |
| CIFAR100 | 19.10 | 31.24 | 19.01 | 32.05 |
| MNIST | 98.20 | 99.27 | 98.25 | 98.69 |
| Fashion MNIST | 88.12 | 89.06 | 88.14 | 88.42 |
| CRX8 | 82.96 | 83.46 | 80.20 | 81.11 |
| Melanoma | 78.68 | 79.36 | 77.23 | 77.85 |

The precision determines the repeatability of obtaining the models' good performance, indicating how well the model can predict the correct class. Table 14 looks at the recall performance of using the positive samples to determine the model's performance in correctly identifying the positives. Figure 6 shows a confusion matrix of the ISIC 2016 testing dataset on MobileNetV2, where the number of false negatives and false positives is lower in classifying benign or malignant conditions.

In Table 14, the recall values are lower than the accuracy and the precision by a slight margin. However, it proves that the model can identify a good proportion of the positives during classification.

**Table 9 InceptionV3 accuracy performance using the selected DKL methods.**

| Dataset | VGG16 | | | |
| --- | --- | --- | --- | --- |
| | Kullback–Leibler methods | | | |
| | Positive | | Negative | |
| | Before samples | After samples | Before samples | After samples |
| Caltech 256 | 96.41 | 97.53 | 95.86 | 96.08 |
| MIT Indoor | 91.05 | 91.29 | 90.36 | 90.58 |
| Stanford Dogs | 90.54 | 92.87 | 91.57 | 92.01 |
| CIFAR10 | 73.14 | 75.24 | 73.08 | 74.56 |
| CIFAR100 | 37.62 | 45.24 | 34.72 | 41.05 |
| MNIST | 99.52 | 99.81 | 99.27 | 99.54 |
| Fashion MNIST | 88.21 | 91.24 | 87.96 | 90.14 |
| CRX8 | 85.38 | 86.44 | 82.30 | 84.50 |
| Melanoma | 84.63 | 85.27 | 82.24 | 82.45 |

**Table 10 DenseNet169 accuracy performance using the selected DKL methods.**

| Dataset | InceptionV3 | | | |
| --- | --- | --- | --- | --- |
| | Kullback–Leibler methods | | | |
| | Positive | | Negative | |
| | Before samples | After samples | Before samples | After samples |
| Caltech 256 | 93.24 | 94.02 | 92.14 | 92.34 |
| MIT Indoor | 89.42 | 90.23 | 87.24 | 88.34 |
| Stanford Dogs | 88.36 | 91.05 | 88.09 | 89.02 |
| CIFAR10 | 77.58 | 78.29 | 78.47 | 79.11 |
| CIFAR100 | 30.59 | 34.12 | 28.98 | 32.21 |
| MNIST | 98.09 | 99.08 | 98.01 | 98.78 |
| Fashion MNIST | 87.53 | 88.24 | 87.45 | 88.03 |
| CRX8 | 80.48 | 81.45 | 78.32 | 79.38 |
| Melanoma | 85.36 | 85.98 | 81.39 | 82.35 |

## Results of proposed methods against commonly used transfer learning methods

The introduced methods of the proposed approach have been compared to the three commonly used methods of the k-1, k-2 and k-3. Table 15 shows that the introduced methods outperform these regular methods.

The combination of the positive $D_{KL}$ in dynamic selected layers and the use of data points below conflated average $D_{KL}$ give an average improvement of 0.87% across all the five pre-trained models, with the DenseNet169 model giving the best improvement of 1.57% and the VGG16 model giving the slightest improvement of 0.06% in comparison to

**Table 11 MobileNetV2 accuracy performance using the selected DKL methods.**

| Dataset | DenseNet169 | | | |
| | Kullback–Leibler methods | | | |
| | Positive | | Negative | |
| | Before samples | After samples | Before samples | After samples |
|---|---|---|---|---|
| Caltech 256 | 91.54 | 92.69 | 90.47 | 90.65 |
| MIT Indoor | 87.28 | 88.90 | 86.88 | 86.82 |
| Stanford Dogs | 86.65 | 87.35 | 84.07 | 84.98 |
| CIFAR10 | 69.18 | 72.49 | 69.05 | 71.56 |
| CIFAR100 | 34.58 | 42.15 | 32.15 | 41.49 |
| MNIST | 99.02 | 99.52 | 98.89 | 98.94 |
| Fashion MNIST | 87.69 | 88.14 | 87.12 | 88.01 |
| CRX8 | 78.24 | 79.61 | 77.36 | 77.58 |
| Melanoma | 81.36 | 82.35 | 79.35 | 81.44 |

**Table 12 Accuracy performance on the approach methods and the standard baselines–CRX8 (homogeneity).**

| Dataset | MobileNetV2 | | | |
| | Kullback–Leibler methods | | | |
| | Positive | | Negative | |
| | Before samples | After samples | Before samples | After samples |
|---|---|---|---|---|
| Caltech 256 | 91.45 | 92.34 | 93.65 | 94.04 |
| MIT Indoor | 86.24 | 87.95 | 86.02 | 86.39 |
| Stanford Dogs | 88.56 | 88.96 | 87.34 | 87.58 |
| CIFAR10 | 64.21 | 66.20 | 64.11 | 65.14 |
| CIFAR100 | 29.99 | 32.41 | 29.38 | 30.04 |
| MNIST | 97.98 | 98.56 | 97.90 | 97.87 |
| Fashion MNIST | 87.35 | 88.69 | 87.04 | 88.96 |
| CRX8 | 77.53 | 79.34 | 67.08 | 70.12 |
| Melanoma | 76.40 | 78.15 | 73.24 | 73.69 |

the standard fine-tuning as seen in Table 15 for the ChestX-ray8 dataset. Among the last-k methods, the combination dramatically improves on the k-3. A similar performance is shown in Fig. 7.

Combining data points below the average $D_{KL}$ and dynamically selected layers gives better accuracy than the commonly used and individually introduced methods. Utilising above-average $D_{KL}$ samples without dynamically selected layers gives the least performance, as shown in Fig. 7. Using below-average samples gives the second-best method. Figure 8 shows the ISIC 2016 dataset sample with and without transfer learning in

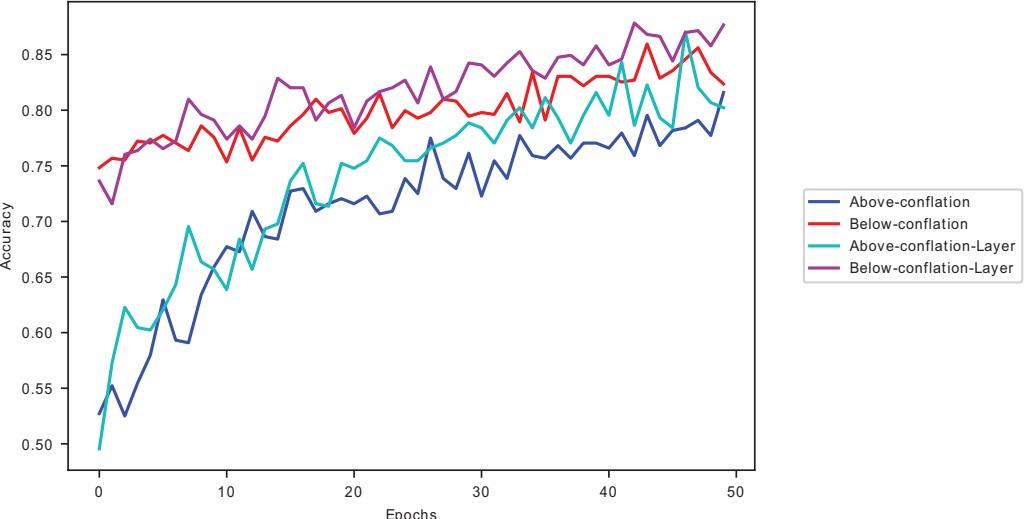

**Figure 4 A plot of the Melanoma dataset on VGG16 with conflated dataset and dynamically selected layers.**

a) VGG16 - Layer 1, channel 1 weights

b) VGG16 - Layer 9, channel 1 weights

**Figure 5 Proposed methods and baseline curves using the Melanoma dataset on VGG16.**

**Table 13 Computational complexity of divergence measure on Caltech256.**

| Dataset | MobileNetV2 | | | |
| --- | --- | --- | --- | --- |
| | Kullback–Leibler methods | | | |
| | Positive | | Negative | |
| | Before samples | After samples | Before samples | After samples |
| Caltech 256 | 90.34 | 91.85 | 93.15 | 93.88 |
| MIT Indoor | 86.02 | 87.36 | 85.85 | 85.74 |
| Stanford Dogs | 89.01 | 90.37 | 86.48 | 86.95 |
| CIFAR10 | 65.23 | 67.92 | 64.06 | 66.24 |
| CIFAR100 | 30.08 | 33.74 | 29.04 | 29.51 |
| MNIST | 98.36 | 98.87 | 96.47 | 96.94 |
| Fashion MNIST | 88.41 | 90.21 | 88.14 | 89.38 |
| CRX8 | 78.10 | 80.39 | 66.47 | 69.54 |
| Melanoma | 77.63 | 79.54 | 73.12 | 74.85 |

**Table 14 MobileNetV2 Recall performance using the selected DKL methods.**

| Dataset | MobileNetV2 | | | |
| --- | --- | --- | --- | --- |
| | Kullback–Leibler methods | | | |
| | Positive | | Negative | |
| | Before samples | After samples | Before samples | After samples |
| Caltech 256 | 87.41 | 89.18 | 86.13 | 88.20 |
| MIT Indoor | 84.38 | 85.47 | 83.69 | 84.37 |
| Stanford Dogs | 87.82 | 89.30 | 83.54 | 85.25 |
| CIFAR10 | 64.29 | 66.85 | 61.24 | 64.97 |
| CIFAR100 | 29.58 | 31.84 | 28.23 | 28.78 |
| MNIST | 96.49 | 97.63 | 93.16 | 94.08 |
| Fashion MNIST | 86.76 | 88.89 | 87.07 | 88.92 |
| CRX8 | 76.81 | 78.14 | 63.45 | 67.83 |
| Melanoma | 75.24 | 76.86 | 71.98 | 72.23 |

the VGG16 pre-trained model. When the same samples are used in transfer learning, it is noted that below-average $D_{KL}$ performs better and takes less time than a typical convolutional neural network (a 12-layer CNN listed in Table 2). The typical CNN takes 50 s longer to train the ISIC 2016 samples.

## Results on computational complexities in proposed methods

The proposed approach complexity is evaluated against four divergence measures: Wasserstein, Hellinger, Jensen–Shannon and Bhattacharya. The results in Table 16 show that the $D_{KL}$ has a good balance of memory and time complexities.

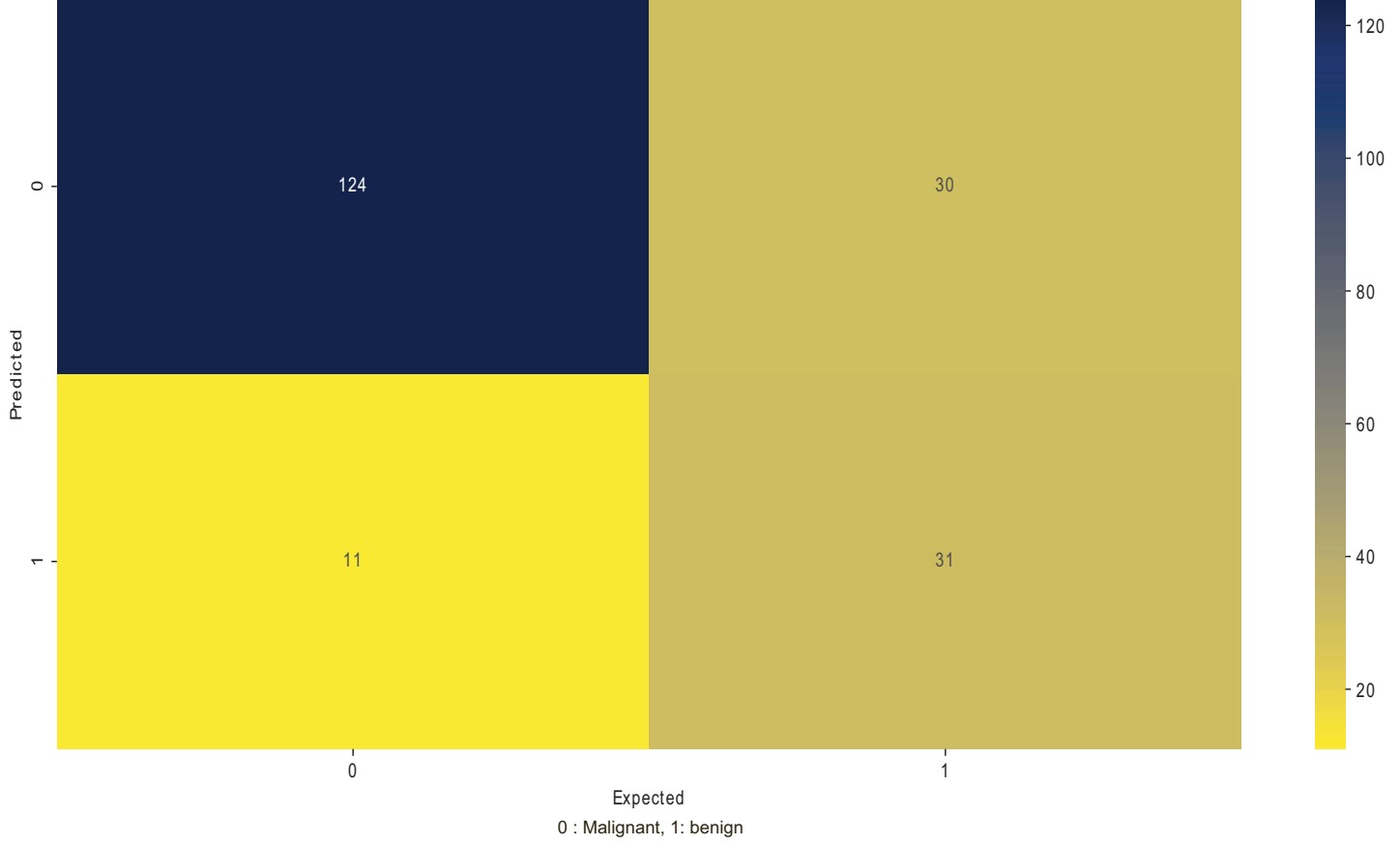

**Figure 6 Divergences validation loss curves on CIFAR10 on ResNet50.**

**Table 15 Accuracy performance on the approach methods and the standard baselines–CRX8 (homogeneity).**

| Method | Model | | | | |
|---|---|---|---|---|---|
| | ResNet50 | VGG16 | InceptionV3 | DenseNet169 | MobileNetV2 |
| Below DKL | 84.34 | 72.14 | 84.15 | 81.26 | 77.25 |
| Above DKL | 83.97 | 68.90 | 81.20 | 78.50 | 72.45 |
| Positive DKL | 82.96 | 85.38 | 80.48 | 78.24 | 77.53 |
| Negative DKL | 80.20 | 82.30 | 78.32 | 77.36 | 67.08 |
| Positive DKL+Above DKL | 83.46 | 86.44 | 81.45 | 79.61 | 79.34 |
| Standard fine-tuning | 83.08 | 86.38 | 80.05 | 78.04 | 78.40 |
| Last k-1 | 80.14 | 82.54 | 78.98 | 77.39 | 78.18 |
| Last k-2 | 82.26 | 84.03 | 79.59 | 77.65 | 79.04 |
| Last k-3 | 82.98 | 84.84 | 79.56 | 77.84 | 79.12 |

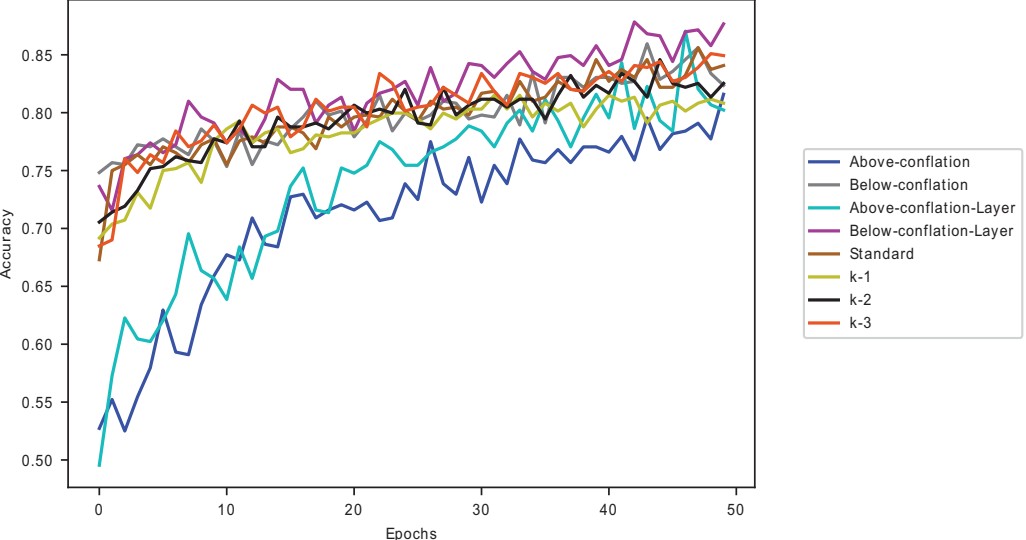

**Figure 7** **Proposed methods and baseline curves using the Melanoma dataset on VGG16.**

The Wasserstein and the Jensen–Shannon have better time complexities but consume more computing resources than the $D_{KL}$. Tables 16 and 17 show that it takes about 70.37 milliseconds (ms) to select a sample and pass it through a selected layer in ResNet50. The complexities of the $D_{KL}$ are better than Hellinger's and Bhattacharyya's, which take 99.63 and 239.14 ms, respectively. According to *Yossi, Carlo & Leonidas (2000)*, the Wasserstein is reported to be more complex than Bhattacharyya, forming a reasonable basis for selecting $D_{KL}$ in this study. Further evaluation of the divergence measures is shown in Fig. 9.

The models' accuracy performance when using $D_{KL}$ conflated dataset samples is closest to Hellinger's conflated dataset samples, as seen in Fig. 9, outranking the other divergences. However, as reported in Tables 16 and 17, Hellinger has a higher computational complexity.

## DISCUSSION

From the results, the performance of the pre-trained models is improved at two levels: selecting the relevant data points and utilising dynamically selected layers. In selecting the relevant data points, GLCM's energy and homogeneity properties have been noted to perform well compared to the other properties: GLCM's correlation and LBP. The contribution of these two to good performance can be attributed to the excellent neighbouring of pixels with similar grey levels. The GLCM's homogeneity causes the low density of the pixels' edges. *Chaves (2021)* and *Mathworks (2023)* note that pixels along the diagonal change smoothly to the ones distant from the main diagonal. The samples with below-average $D_{KL}$ contain values closer to each other, with many adjacent pixels having similar values. Pixels of lower values are far from the diagonal.

The performance of the GLCM's properties is better than the LBP textural descriptor in many instances. This performance can be attributed to better uniformity and simplicity in

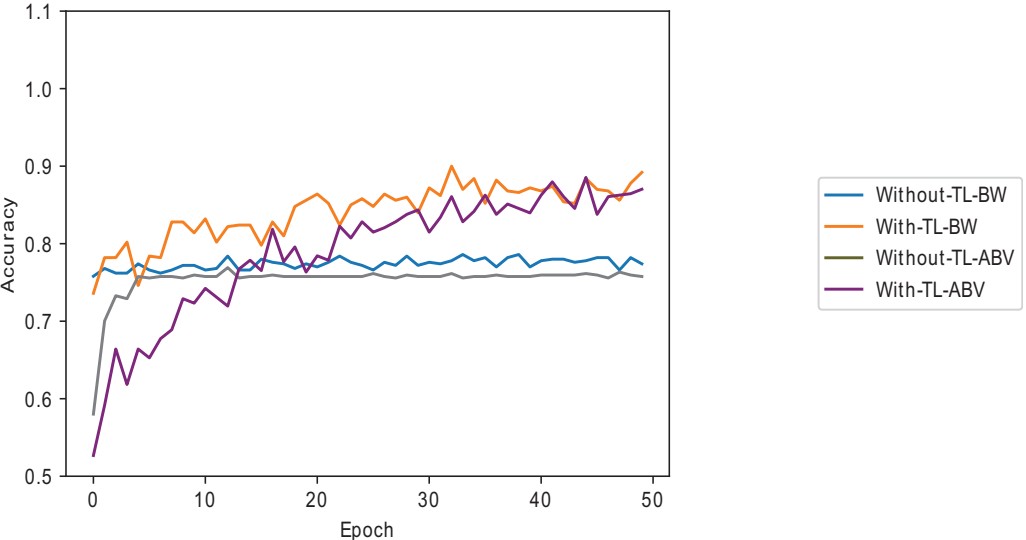

**Figure 8 Melanoma dataset on the VGG16 model with and without transfer learning.**

**Table 16 Computational complexity of divergence measure on Caltech256—sample selection.**

| Model | Divergence measures | | | | | | | | | |
|---|---|---|---|---|---|---|---|---|---|---|
| | Kullback–Leibler | | Wasserstein | | Hellinger | | Jensen–Shannon | | Bhattacharya | |
| | Time | Mem | Time | Mem | Time | Mem | Time | Mem | Time | Mem |
| ResNet50 | 37.48 | 104,137 | 13.96 | 30,297 | 51.16 | 376,325 | 6.51 | 238,501 | 118.78 | 1,159,044 |
| InceptionV3 | 36.83 | 28,151 | 18.35 | 44,451 | 67.74 | 672,025 | 7.10 | 414,769 | 164.84 | 1,639,686 |
| MobileNetV2 | 14.57 | 40,900 | 3.654 | 39,695 | 4.67 | 40,527 | 4.12 | 40,475 | 3.668 | 37,805 |
| VGG16 | 5.34 | 25,315 | 3.75 | 269,011 | 3.98 | 384,880 | 3.66 | 187,836 | 3.887 | 187,836 |
| DenseNet169 | 908.15 | 320,543 | 1,174.06 | 632,111 | 1,768.06 | 1,262,959 | 78.93 | 787,334 | 1,836.14 | 1,862,471 |

the texture, as noted by *Park et al. (2011)*, who describes that energy and homogeneity give better performance, a similar trend noted in this work. The performance of GLCM properties to LBP has been cited in the literature to outperform LBP due to its less grey-level feature discrimination, as noted by *Changwei et al. (2020)* and *Nurhaida, Manurung & Arymurthy (2012)*. Its performance in CNNs is also reported by *Tan et al. (2020)*, indicating that the use of GLCM properties can aid in improving CNNs' performance, especially in cases of inadequate data—a use case for transfer learning. In using below-average $D_{KL}$ samples, the selected data points utilise their lower informational differences to the source samples adapting the target task.

At the dynamic selection of layers, it is noted that layers with lower positively signed weights to each other give higher performance than the negative ones. The positive weights in a layer are considered excitatory, as *Najafi et al. (2020)* noted, with the ability to select stimulating features during the model's training. As noted in Fig. 5, the weights of the first channel of the first layer would start the convolutional process with a higher magnitude

**Table 17 Complexity comparison to other divergence measures—layer selection.**

| Model | Kullback–Leibler | | Wasserstein | | Hellinger | | Jensen–Shannon | | Bhattacharya | |
|---|---|---|---|---|---|---|---|---|---|---|
| | Time | Mem | Time | Mem | Time | Mem | Time | Mem | Time | Mem |
| ResNet50 | 32.89 | 85,444 | 15.86 | 1,811,638 | 48.47 | 370,086 | 5.25 | 232,424 | 120.36 | 1,450,276 |
| InceptionV3 | 33.84 | 90,332 | 17.89 | 3,224,079 | 66.91 | 656,161 | 5.35 | 406,236 | 194.11 | 1,627,010 |
| MobileNetV2 | 1.92 | 40,323 | 0.84 | 916,694 | 2.68 | 193,214 | 0.51 | 129,121 | 6.20 | 716,486 |
| VGG16 | 0.11 | 13,106 | 0.08 | 255,810 | 0.18 | 51,806 | 0.05 | 32,183 | 0.22 | 176,199 |
| DenseNet169 | 56.25 | 125,025 | 508.47 | 6,202,006 | 96.24 | 4,212,014 | 110.69 | 797,576 | 219.56 | 3,501,420 |

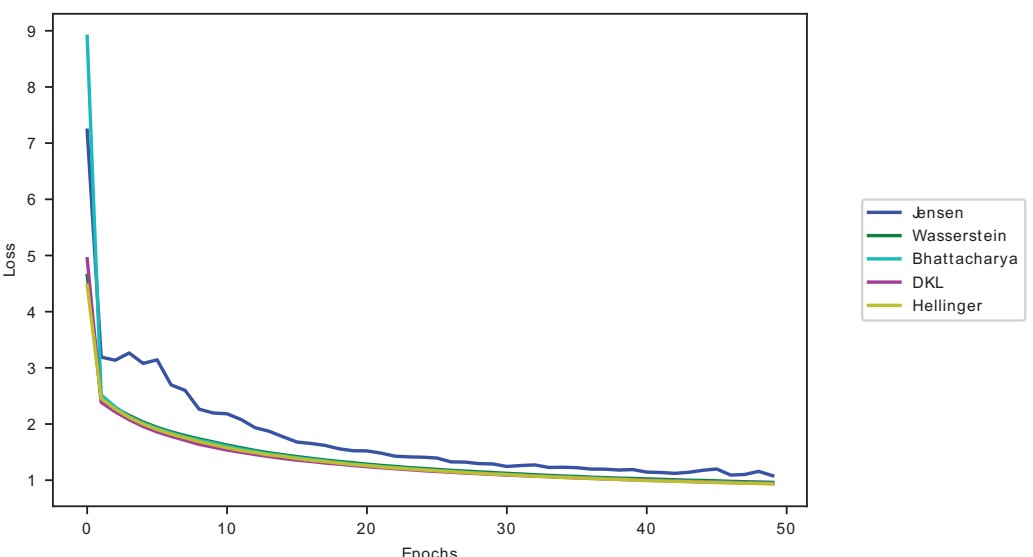

**Figure 9 Divergences validation loss curves on CIFAR10 on ResNet50.**

(the RGB values of colours closer to white would be near 255) towards the divergence than the counterpart filter in the first channel of layer 9. This property leads to better convergence and faster training, as *Delaurentis & Dickey (1994)* noted. In the case of using negative weights, the model cannot descend well hence the use of positive weights, as noted by *Shamsuddin, Ibrahim & Ramadhena (2013)*. The use of weights is essential to the model's training, where the convergence happens despite the use of the other parameters since its change has an effect in reaching the global minima. The change in weight sign affects the magnitude change and consequent change in the direction of the descent.

Sidani gives more intuitive reasoning on the effects of positive weights on the training process by noting that an increase in the weight of the previous and the current derivative stabilises the network more, leading to quicker descent. Therefore the positive weights in the training process can correct the back-propagation errors leading to a better path to the global minima. The negative weights $D_{KL}$ method is also noted to improve the use of better samples. This improvement by the negative weights (inhibitory) is because they still

stabilise the training process, especially in cases of exploding gradient. This process slows the learning process but ensures the model can capture the features well enough. However, the positive weights have the upper hand in directing the model to the global minima.

It is also noted that heavy pre-trained models like DenseNet169 with many layers (as shown in Table 2) and datasets with many classes like CIFAR-100 give a lower performance. These could be a result of parameter complexity requiring more training. This behaviour has been reported by *Vrbančič & Podgorelec (2020)* in the DEFT method. The heavy DenseNet169 model also gives higher complexity, as Tables 16 and 17 note.

The selected $D_{KL}$ methods have low-medium time complexity compared to the other divergences despite the lower time complexity by Jensen and Wasserstein and Hellinger's better training loss curve to the $D_{KL}$. However, comparing the $D_{KL}$ memory complexity and the combined complexities of the other divergences, as seen in Tables 16 and 17, can unfold into an extensive complexity which guided its selection for use in this study.

## CONCLUSION

This article introduces the conflation of features in selecting quality data points in a target domain dataset and using weights in the dynamic selection of fine-tuning layers to improve the transfer learning process. The enhanced model demonstrates that using the correct data points and suitable layers can improve the performance of a pre-trained model to the commonly used transfer learning methods. The model has been evaluated on five pre-trained models and nine datasets. The results demonstrate the divergence between data points and layers, showing how transfer learning adaptation is affected by information divergence at the data and layer levels. However, the approach has a higher time complexity than the commonly used methods due to adding the extra step in dynamic layer selection. The approach gives a better method in cases of inadequate data reducing cases of trial and error in selecting the right data points and layers for fine-tuning.

Future work can be extended into other architectures apart from CNNs to understand further the importance of divergence in data samples and the models' layers.

### Funding
The authors received no funding for this work.

### Competing Interests
The authors declare that they have no competing interests.

### Author Contributions
- Raphael Ngigi Wanjiku conceived and designed the experiments, performed the experiments, analyzed the data, performed the computation work, prepared figures and/ or tables, authored or reviewed drafts of the article, and approved the final draft.
- Lawrence Nderu conceived and designed the experiments, performed the experiments, analyzed the data, prepared figures and/or tables, authored or reviewed drafts of the article, and approved the final draft.

- Michael Kimwele conceived and designed the experiments, performed the experiments, authored or reviewed drafts of the article, and approved the final draft.

## Data Availability

The public datasets are available at:

1. CIFAR10—https://www.cs.toronto.edu/~kriz/cifar.html

A zipped repository is available at Kaggle: https://www.kaggle.com/datasets/oxcdcd/cifar10

2. CIFAR100—https://www.cs.toronto.edu/~kriz/cifar.html

A zipped repository is available at Kaggle: https://www.kaggle.com/datasets/aymenboulila2/cifar100

3. MNIST—https://www.kaggle.com/datasets/jidhumohan/mnist-png

4. Fashion Mnist—https://github.com/zalandoresearch/fashion-mnist/blob/master/LICENSE

A zipped repository is available at Kaggle: https://www.kaggle.com/datasets/zalando-research/fashionmnist

5. Caltech 256—https://data.caltech.edu/records/nyy15-4j048

6. Stanford Dogs 120—http://vision.stanford.edu/aditya86/ImageNetDogs/main.html

7. MIT Indoor—https://web.mit.edu/torralba/www/indoor.html

8. ISIC Melanoma—https://challenge2020.isic-archive.com/

9. Chest Xray8—https://www.kaggle.com/datasets/nih-chest-xrays/data

The code for the proposed work is available at GitHub and Zenodo:

- https://github.com/geekhack/conflation_dynamic_layers_TL/tree/v1.0.0

- Rafael Wanjiku. (2023). geekhack/conflation_dynamic_layers_TL: v1.0.0 (v1.0.0). Zenodo. https://doi.org/10.5281/zenodo.8048467

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
