# Peer review of "Improved transfer learning using textural features conflation and dynamically fine-tuned layers"

_PeerJ Computer Science, doi:10.7717/peerj-cs.1601_

## Round 0.1 · original submission · Major Revisions

Please follow review recommendations.

Reviewer 1 ·

Basic reporting

The authors proposed a new transfer learning method using textural feature divergence and layers with more positive weights. They reported that the proposed method yielded better accuracy. However, upon going through the manuscript, I have a few queries and suggestions, which are given below:

1. I feel that the grammar is incorrect in many places, and the messages are not presented correctly. For instance, the sentence, "With the emergence of ImageNet, CNNs changed the landscape of artificial intelligence through the use of many hidden layers in neural networks," could be improved grammatically and rephrased for clarity.

2. The literature section is shallow, and I request the authors to cite more recent papers using transfer learning in different areas, such as biomedical imaging and diagnosis, in the literature.

Experimental design

3. I request the authors to redraw Fig. 1, "Conceptual framework of the proposed method," by referring to and citing the paper: https://doi.org/10.1016/j.bea.2022.100069

Validity of the findings

4. In the conclusion section, the authors state that "the approach has a higher complexity than the standard baselines." However, I cannot find any comparison of the models in terms of learning parameters or depth of the models considered in any of the tables. It would be better to include this information as well.

Additional comments

The problem identified is interesting, and the quality of the manuscript can be further improved by incorporating the above suggestions.



All the best.

·

Basic reporting

The motivation of the proposed wok is not clear.

Experimental design

The proposed work lacks novelty.

Validity of the findings

The proposed work lacks novelty as the outcome of transfer learning is well established in the literature.

Reviewer 3 ·

Basic reporting

This work reduces the knowledge lost during transfer learning by utilising the closest textural features of the target similar datasets and layers with the least divergence based on the pre-trained model weights. The work is demonstrated using a variety of datasets, with an excellent introduction section, literature review, methodology and results and discussion sections. There is a need for a comprehensive grammar check, but the presentation is straightforward.

Experimental design

It is beneficial to demonstrate the outcomes of deep learning networks without transfer learning. Precision, recall, and accuracy can be used to describe the classification accuracy.

Validity of the findings

A layer-weight visualisation of the approach can help the reader comprehend the efficacy of the proposed method.
The confusion matrix can be utilised to validate the results appropriately.

---

## Round 0.2 · Major Revisions

Please address changes and resubmit.

Reviewer 1 ·

Basic reporting

The overall quality of the revised manuscript has improved after the revision, but there are a few places where the authors need to make some minor corrections before final publication.

Experimental design

1. I believe "ISIC Melanoma" is not the correct name for the dataset. Please refer to the latest published work that uses the ISIC skin cancer database: https://doi.org/10.1016/j.dajour.2023.100278.

2. I couldn't find any references to the databases used in the reference section. The authors need to include them before publishing.

3. The authors have made substantial improvements to the revised manuscript, particularly in terms of the English language. However, I suggest that some subsections, table titles, and figure captions could be more meaningful. For example, the subsection titled "Standard Baselines" does not convey a clear meaning to the readers.

4. The sentence in the Conclusion, "This paper combines target-selected data points and dynamically selected fine-tuning layers," does not effectively convey the authors' intended message. Please rephrase it in a more meaningful way.

Validity of the findings

The findings of this research are valid and valuable for the research community.

---

## Round 0.3 · accepted · Accept

The authors have addressed all of the reviewers' comments.